

# Data processing choices can affect findings in differential methylation analyses: an investigation using data from the LIMIT RCT

Jennie Louise[1,2], Andrea R. Deussen[1] and Jodie M. Dodd[1,3]

[1] Discipline of Obstetrics & Gynaecology and The Robinson Research Institute, The University of Adelaide, Adelaide, Australia

[2] Adelaide Health Technology Asseessment, The University of Adelaide, Adelaide, Australia

[3] Department of Perinatal Medicine, Women's and Babies Division, The Women's and Children's Hospital, Adelaide, South Australia, Australia

Corresponding author
Jennie Louise,
jennie.louise@adelaide.edu.au

## ABSTRACT

**Objective**. A wide array of methods exist for processing and analysing DNA methylation data. We aimed to perform a systematic comparison of the behaviour of these methods, using cord blood DNAm from the LIMIT RCT, in relation to detecting hypothesised effects of interest (intervention and pre-pregnancy maternal BMI) as well as effects known to be spurious, and known to be present.

**Methods**. DNAm data, from 645 cord blood samples analysed using Illumina 450K BeadChip arrays, were normalised using three different methods (with probe filtering undertaken pre- or post- normalisation). Batch effects were handled with a supervised algorithm, an unsupervised algorithm, or adjustment in the analysis model. Analysis was undertaken with and without adjustment for estimated cell type proportions. The effects estimated included intervention and BMI (effects of interest in the original study), infant sex and randomly assigned groups. Data processing and analysis methods were compared in relation to number and identity of differentially methylated probes, rankings of probes by p value and log-fold-change, and distributions of *p* values and log-fold-change estimates.

**Results**. There were differences corresponding to each of the processing and analysis choices. Importantly, some combinations of data processing choices resulted in a substantial number of spurious 'significant' findings. We recommend greater emphasis on replication and greater use of sensitivity analyses.

# INTRODUCTION AND BACKGROUND

With the advent of high-throughput assays, epigenome-wide DNA methylation studies have become more popular, and researchers are now investigating the effects on DNA methylation (DNAm) of a wide range of environmental exposures and physiological conditions, with particular interest in the contribution of epigenetic mechanisms such as DNAm to the early life origins of health and disease. The ability to perform EWAS is particularly useful in relation to conditions where associated differences in DNAm

are likely to be fairly modest (*Marabita et al., 2013*). However, DNAm data—as with high-dimensional 'omics' data generally—requires substantial pre-processing prior to analysis, including probe and sample filtering, normalisation to remove variation due to technological factors, and correction for other factors which may confound effects of interest, such as batch effects or differences in cell type proportions between samples. Numerous methods exist to perform these processing steps, and many articles have been published which provide useful guidance for the use of analysis pipelines (*Marabita et al., 2013*; *Yousefi et al., 2013*; *Lehne et al., 2015*; *Morris & Beck, 2015*; *Maksimovic, Phipson & Oshlack, 2017*), or comparing some alternatives for individual steps in the overall processing pipeline, including probe filtering (*Heiss & Just, 2019*), normalisation (*Wang et al., 2012*; *Wang et al., 2015*; *Fortin et al., 2014*; *Wu et al., 2014*; *Hicks & Irizarry, 2015*), or correction for/avoidance of batch effects. These have led to some general conclusions regarding the need to account for batch effects, the importance of correcting for estimated cell type proportion, and perhaps the greater suitability of within-array normalisation methods compared to between-array methods when global methylation differences are not expected (*Maksimovic, Phipson & Oshlack, 2017*), but there is no clear overall consensus on the best processing or analysis approach (*Price & Robinson, 2018*; *Zindler et al., 2020*), or of the overall advantages and disadvantages of different combinations of processing choices.

We recently investigated the effect of an antenatal diet and lifestyle intervention, and of maternal early pregnancy BMI, on neonatal cord blood DNA methylation in infants of mothers who were overweight or obese in early pregnancy (*Louise et al., 2022*). We were unable to replicate findings from previous studies which reported a range of loci to be significantly differentially methylated in relation to maternal BMI or diet and lifestyle in pregnancy (*Gemma et al., 2009*; *Sharp et al., 2015*; *Sharp et al., 2017*; *Thakali et al., 2017*; *Hjort et al., 2018*) and indeed did not find any significant differences in methylation corresponding to either BMI or intervention effects. We were aware of literature suggesting that use of supervised batch-correction algorithms may produce spurious findings (*Nygaard, Rødland & Hovig, 2016a*; *Price & Robinson, 2018*; *Zindler et al., 2020*), and that the number of statistically significant findings may differ according to normalisation method (*Wu et al., 2014*), adjustment for estimated cell type proportion (*Sharp et al., 2017*) or stringency of type I error control (*Wu et al., 2014*), which led to the hypothesis that the discrepancy in findings may be due in part to differences in data processing and analysis methods. Following the common practice in clinical trials of conducting sensitivity analyses to assess robustness of results to various assumptions and decisions (*Thabane et al., 2013*), we performed several re-analyses with different normalisation methods, methods for batch effect handling, type I error control, and presence *vs* absence of adjustment for estimated cell type proportions. This confirmed that our findings indeed differed under different data-processing and analysis choices.

While previous studies comparing different methods have also produced different findings, these have tended to consider only one element of the processing and analysis pipeline (such as normalisation, or batch correction) in isolation. In addition, they have tended to concentrate on the tendency for some methods to produce results which are likely to be spurious (false positives), while often being unable to definitively confirm

this due to the lack of known truth regarding the presence and magnitude of differential methylation effects.

We therefore set out to investigate the impact of different data-processing choices in a more systematic way, looking at the effect of combinations of data processing choices on findings specifically regarding statistically significant differentially methylated probes (DMPs), and of the behaviour of these combinations in cases where effects are known to be either absent or present, as well as their behaviour in relation to our effects of interest (maternal BMI and lifestyle intervention). We were able to create a scenario in which effects were known to be absent by randomly assigning samples to groupings. We could not similarly ensure a scenario where effects were known to be present (as the truth regarding the existence, location and magnitude of any effects in our samples is not known); however we investigated effects of infant sex as a rough proxy, since infant sex is known to have substantial effects on DNAm which can be detected by the 450K array (*Yousefi et al., 2015*).

## DATA AND METHODS

### The LIMIT randomised controlled trial

The LIMIT study was a randomised, controlled trial of an antenatal diet and lifestyle intervention for women with early pregnancy BMI $\geq 25.0$ kg/m$^2$. The study, and its primary and main secondary outcomes, have been extensively reported elsewhere (*Dodd et al., 2014b*). Inclusion criteria were early pregnancy BMI $\geq 25.0$ kg/m$^2$ and pregnancy between $10^{+0}$ and $20^{+0}$ weeks' gestation, with exclusion criteria of multiple gestation or previously existing diabetes. The study randomised 2,212 women in total to one of two groups: a comprehensive diet and lifestyle intervention (Lifestyle Advice; $n = 1,108$) or antenatal care delivered according to local guidelines (Standard Care; $n = 1,104$) which did not include information on diet or physical activity. The study was reviewed by the ethics committee of each participating institution including the Women's and Children's Health Network Human Research Ethics Committee (1839 & 2051), the Central and Northern Adelaide Health Network Human Research Ethics Committee (2008033) and the Southern Adelaide Local Health Network Human Research Ethics Committee (128/08). Informed written consent was obtained for all participants to participate in the LIMIT study, and additional written consent was obtained to collect samples of umbilical cord blood at delivery for the purposes of gene expression research related to weight and to the diet and lifestyle intervention.

The primary outcome of the LIMIT study was birth of a large for gestational age (LGA) infant. There were no significant differences observed between the groups in relation to this outcome; however, a significantly lower incidence of high birthweight (>4 kg) was observed in the Lifestyle Advice group, with a Relative Risk of 0.82 (95% CI: 0.68, 0.99, $p = 0.04$). Additionally, measures of diet quality and physical activity were improved in women in the Lifestyle Advice group compared to those in the Standard Care group (*Dodd et al., 2014a*).

As previously outlined in the companion paper (*Louise et al., 2022*), cord blood DNA for a range of secondary studies was collected at the time of birth from consenting participants,

and was frozen as whole blood preserved with EDTA. Funding was available to perform DNA methylation analysis for a total of 649 samples, which were randomly selected from the total number of available samples, balanced between the Lifestyle Advice and Standard Care groups (Table S1). After DNA extraction, genome-wide DNA methylation was performed using the Illumina Infinium HumanMethylation 450K Bead-Chip array. Results were supplied as raw probe intensities (IDAT files).

For the additional analyses investigating known spurious effects, artificial (fake) groups were created by assigning samples based on random draws from binomial distributions. The first grouping ('Tortoiseshell' *vs* 'Tabby') was generated using a binomial distribution with 50% probability of assignment to each group. The second grouping ('Long'- *vs* 'Short-Haired') was created to mimic stratified randomisation as well as unequal proportions in each group: within each level of the first fake group, samples were assigned to Long-Haired with 40% probability and Short-Haired with 60% probability.

All data processing and analyses were undertaken using R version 4.0 (*R Core Team, 2018*).

## Probe and sample filtering

The *minfi* package (*Aryee et al., 2014*) was used to read in the raw *idats* (without normalisation), and to calculate both probe-wise and sample-wise detection $p$ values. Samples were identified as faulty if they had a detection $p$-value $\geq 0.05$. A total of 13 such samples were excluded; however these were due to a known chip failure, and had subsequently been rerun. A further four samples were excluded because the correct corresponding study identifier could not be ascertained, leaving 645 samples for analysis.

Probes were filtered using multiple criteria. Firstly, probes were excluded if they had a detection $p$-value $\geq 0.001$ in more than 25% of the 645 samples, indicating that their signal could not be accurately detected for a large proportion of samples (*Dedeurwaerder et al., 2014*; *Maksimovic, Phipson & Oshlack, 2017*). Secondly, probes were excluded if they were on a list of those previously identified as cross-reactive (*Chen et al., 2013*); *i.e.,* there was a high probability they may hybridize to locations on the genome different to those for which the probe was designed (*Dedeurwaerder et al., 2014*; *Naeem et al., 2014*). Thirdly, probes with an identified SNP within 3 nucleotides of the CpG site and minor allele frequency >1%, and probes on the X and Y chromosomes were excluded. This was done in order to avoid spurious methylation 'differences' due either to SNPs within the CpG targets, or due to X and Y chromosomes (*Dedeurwaerder et al., 2014*; *Naeem et al., 2014*). Filtering of cross-reactive probes, probes with a nearby SNP and probes on the X and Y chromosomes was performed using the *DMRCate* package (*Peters et al., 2015*). This left 426,572 probes available for analysis.

Probe filtering was performed either after normalisation (post-filtered) or prior to normalisation (pre-filtered). The one exception for pre-filtering was when normalising using the BMIQ method, where probes on the X and Y chromosome were retained as this was required in order for the function to run.

## Normalisation

Normalisation involves making changes to the raw data in order to remove artifactual variation. In the case of Illumina 450K BeadChip arrays, this requires correcting for the presence of two different probe types. Infinium I probes use the same colour signal for methylated and unmethylated CpG and are often used for regions of high CpG density, while Infinium II probes use different colours to differentiate between methylated and unmethylated states (*Pidsley et al., 2013*; *Wang et al., 2015*). Normalisation is performed on $\beta$ values, or the ratio of methylated to total intensity, defined as $\frac{M}{M+U+offset}$. Here, M is the methylated intensity and U is the unmethylated intensity; the offset is a constant added to regularize the $\beta$ value where both methylated and unmethylated intensities are low. The distribution of $\beta$ values is bimodal, with peaks corresponding to methylated and unmethylated states, but the distribution of Infinium II probes differs from that of Infinium I, being more compressed towards 0.5 (*Pidsley et al., 2013*) and hence having a smaller dynamic range (*Teschendorff et al., 2013*; *Dedeurwaerder et al., 2014*).

Numerous methods exist for normalising Illumina BeadChip array data, but there is little consensus or guidance on which should be employed in a given context. The main advice is that between-array methods, which normalise across samples, are preferable when global differences between samples are expected, while within-array methods, which normalise probes within each sample, are better suited to effects in which the majority of genes will not be differentially expressed. (*Maksimovic, Phipson & Oshlack, 2017*). The latter is the context in which many EWAS studies, including the present one, are conducted; as noted above, only modest differences in a small proportion of genes are expected for most early-life exposures. The methods chosen for the present investigation have all been used in the context of studies such as this: Categorical-Subset Quantile Normalisation (SQN) (*Wu et al., 2014*; *Wang et al., 2015*), Beta-Mixture Quantile Normalisation (BMIQ) (*Teschendorff et al., 2013*), and Subset-Quantile Within-Array Normalisation (SWAN) (*Maksimovic, Gordon & Oshlack, 2012*). While numerous other methods exist, a comparison of all available normalisation methods was beyond the scope of this paper. Further details on the methods are given in the Supplemental Information.

Both Subset Quantile Normalisation and Subset-Within-Array-Normalisation were performed using functions in the *minfi* package (*preprocessQuantile* and *preprocessSWAN* respectively), on raw intensity data. Beta-Mixture Quantile normalisation was performed using the *champ.norm* function in the *ChAMP* package after converting intensities to $\beta$ values.

## Batch effects

Batch effects arise when samples are processed in separate groups, creating unwanted variation due, for example, to different reagents, different plates or different scanner settings (*Morris & Beck, 2015*; *Nygaard, Rødland & Hovig, 2016a*; *Price & Robinson, 2018*).

There are 12 Illumina 450K arrays (samples) per chip (this is reduced to eight arrays per chip for the more recent 850K array); thus most studies involving large numbers of samples must be run on multiple chips. This introduces extra variability to the data, and may also confound the actual effects of interest, if samples from different groups are not

evenly distributed between the batches. These effects must be accounted for in order to obtain valid estimates of the effects of interest.

Unlike probe filtering and normalisation, batch effects can be handled at the analysis stage, by adjusting for batch in the analysis model. However, it is also common to address batch effects at the data-processing stage, using a batch-correction algorithm, with the resulting data considered to be free of batch effects (*Nygaard, Rødland & Hovig, 2016a*). The ComBat algorithm has been widely used and considered the most effective method (*Zindler et al., 2020*) of removing batch effects in DNAm data; it has been incorporated into various analysis pipelines. Until recently, ComBat could be implemented only as a supervised method, in which the biological factors of interest had to be specified along with the batch variable (*Price & Robinson, 2018*; *Fortin, Triche & Hansen, 2016*); it can now also be implemented as an unsupervised method, in which only the batch variable is specified.

For each of the normalised datasets (*i.e.,* SQN, BMIQ and SWAN normalised datasets, each with probes filtered either before or after normalisation), we handled batch effects in three ways: firstly, by adjusting for batch in the analysis model (BatchAdjust); secondly, implementing the supervised ComBat algorithm (sCB); and thirdly, implementing the unsupervised ComBat algorithm (uCB). For the supervised ComBat algorithm, it was necessary to run the process twice: once with the effects of interest specified as maternal early pregnancy BMI, antenatal intervention group, and their interaction; and again with the effects of interest specified as Fake Group 1, Fake Group 2, their interaction, and infant sex.

## Cell type proportions

Cord blood, like whole blood, contains a mixture of different cell types, which have different DNA methylation profiles.(*Jaffe & Irizarry, 2014*; *Teschendorff & Zheng, 2017*) If samples differ in the proportion of these different cell types, this may confound effects of interest, either hiding true differences in DNAm, or giving rise to spurious differences. Most studies of the effect of BMI, lifestyle interventions, or similar factors on cord blood DNA methylation have not attempted (or have not documented an attempt) to correct for potential differences in cell type composition, perhaps because reference profiles for cord blood were not available until more recently (*Bakulski et al., 2016*), and the mix of cell types and DNAm profiles may differ in cord blood compared to whole blood, making it inappropriate to apply reference profiles from whole blood to cord blood data (*Cardenas et al., 2016*).

We estimated the proportion of B cells, CD4+T, CD8+T, granulocytes, monocytes, natural killer, and nucleated RBCs in the raw data using the *estimateCellCounts()* function in the *minfi* package, with the cord blood reference panel. The estimated proportions were then added to the metadata for use as adjustment variables in the analyses. We then undertook analyses either adjusted or not adjusted for estimated cell type proportion.

Figure 1 depicts the combinations of data-processing and analysis choices that were undertaken. In brief, there were six normalised datasets (three different normalisation methods, with probe filtering performed before normalisation or after normalisation). These datasets were either used immediately for analysis, or processed using the ComBat
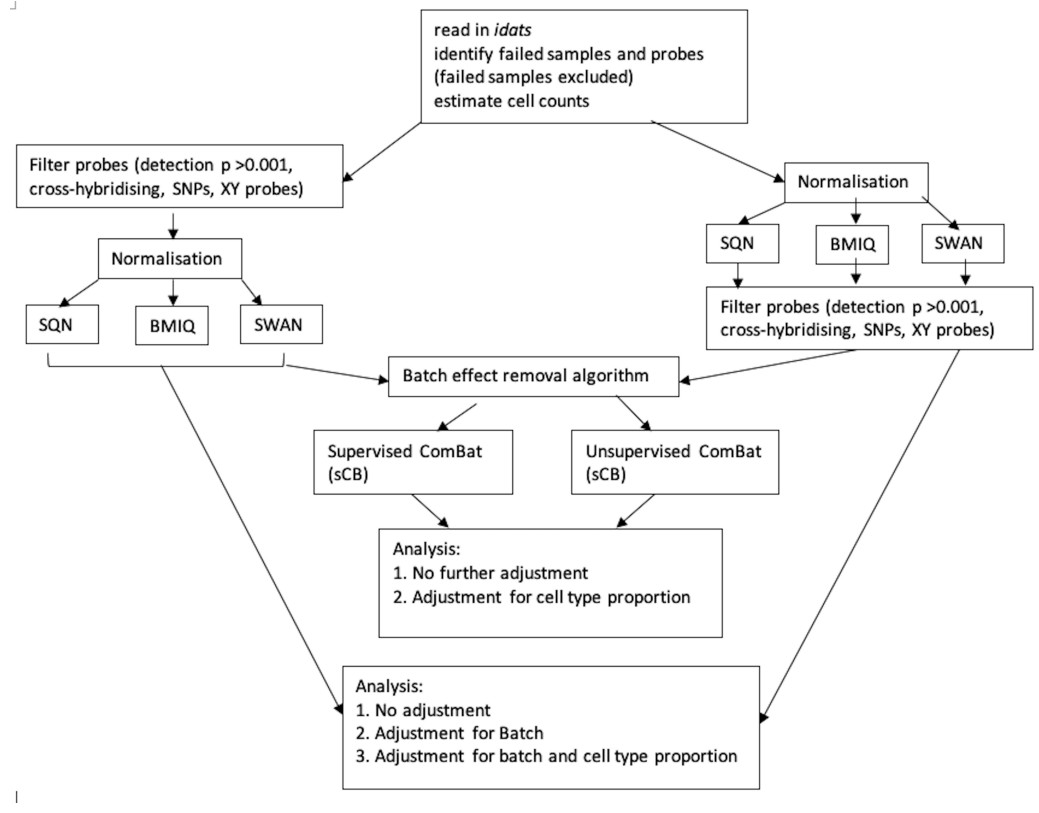

**Figure 1  Flowchart of data processing and analysis.** Combinations of data-processing and analysis choices, consisting of six normalised datasets (SQN, BMIQ or SWAN, with probe filering before or afterwards), use or non-use of ComBat processing (supervised or unsupervised), and analysis with either an unadjusted model, a model adjusted for batch, or a model adjusted for batch and cell type proportion.

algorithm (in both supervised and unsupervised form) prior to analysis. Non-ComBat-processed data were analysed with three different models: an unadjusted model (containing only the effects of interest), a model adjusted for batch, and a model adjusted for batch and estimated cell type proportions. ComBat-processed data were analysed with two different models: one containing no other adjustment variables (but assumed to be pre-adjusted for batch), and one adjusted for cell type proportion.

## Statistical analysis

Differential methylation was investigated probe-wise using linear models with empirical Bayes variance correction as implemented in the *limma* package (*Ritchie et al., 2015*; *Smyth, 2005*). For effects of BMI and intervention, models specified BMI (as continuous and mean-centred), intervention (Lifestyle Advice *vs* Standard Care), and their interaction. Contrasts were specified to estimate the effect of the intervention and of maternal BMI. Because of the presence of the intervention-by-BMI interaction term, this required specification of the BMI values at which the intervention effect was to be estimated, and the intervention groups in which the effect of BMI was to be estimated. For estimating intervention effects, we chose the mean BMI of the cohort (*i.e.*, value of 0 for the mean-centred variable, corresponding

to an actual BMI of approximately 33 kg/m$^2$), and at 5 kg/m$^2$ above the mean (a value of 5, corresponding to an actual BMI of approximately 38 kg/m$^2$. For the effect of BMI, we estimated the effect of an increase of 5 kg/m$^2$ in BMI in each of the intervention groups (Standard Care, Lifestyle Advice) respectively. For effects of fake groups and infant sex, the models specified sex (Female *vs* Male), Fake Group 1 (Tortoiseshell *vs* Tabby), Fake Group 2 (Long-Haired *vs* Short-Haired), and their interaction. Contrasts were specified for infant sex, and for the effect of each fake group separately within levels of the other fake group (*i.e.,* effect of Tortoiseshell in Long-Haired and in Short-Haired; and effect of Short-Haired in Tortoiseshell and Tabby). The model matrix and contrast matrices are shown in the Table S4.

For each contrast in each model, the number and identity (where applicable) of any differentially methylated probes (DMPs) were obtained. For detection of DMPs, *limma*'s default method of multiple-comparisons correction (Benjamini–Hochberg) and default alpha of 0.05 was used; this method controls the false discovery rate, or the proportion of statistically significant results not corresponding to true effects. Where DMPs were obtained, a comparison was made using the Holm method (retaining alpha of 0.05), which controls the Family-Wise-Error Rate (the probability that at least one statistically significant result does not correspond to a true effect). The Holm method can be considered more stringent than Benjamini–Hochberg, but is less stringent than Bonferroni correction, which is known to be too conservative even outside the context of high-dimensional data and is therefore not generally appropriate for EWAS studies (and has not been used in other studies investigating cord blood DNAm in relation to maternal BMI or diet and lifestyle). The full set of *p* values and estimated log-fold-changes (for all 426,572 probes) corresponding to each contrast were also obtained, in order to compare probe rankings and overall distributions. To make the comparison more tractable, probe rankings were investigated using only those probes ranked in the top 10 (*i.e.,* the probes with the smallest *p* value, or largest estimated logFC, in a given model).

The findings from different data-processing choices were then compared along five dimensions:

1. Number and identity of differentially methylated probes (DMPs); for infant sex, the direction of differential methylation ('down', corresponding to negative t-statistics or lower methylation in females, *versus* 'up', corresponding to higher methylation in females) was also examined. For BMI and intervention effects, the number and identity of statistically significant DMPs allows us to see differences in detection of effects, and whether the different analysis pipelines produce consistent results regarding the identity of any DMPs, though the truth is not known. For the fake groupings, the number of statistically significant DMPs is an indication of the tendency to produce spurious findings. For infant sex, while we do not know the actual number and identity of truly differentially methylated sites, differences in the number and identity of DMPs demonstrate that there must be either false positives or false negatives.

2. Consistency of rankings by *p* value for 'top 10' probes. This gives an indication of whether different methods will give the same results for the probes with the largest differences.

3. Consistency of rankings by logFC for 'top 10' probes, as well as the consistency of the logFC estimates. This gives an indication of whether the estimates of effect are similar between methods.

4. Overall distribution of *p* values. Under the null hypothesis of no effect, *p* values should have a uniform distribution between 0 and 1; if effects are present, there will be more *p* values at the lower end of the distribution, the extent of which will depend on how many DMPs there are and the strength of the effects.

5. Overall distribution of logFC estimates. Under the null hypothesis of no effect, logFC estimates should be roughly normally distributed around 0. If effects are present, there will be more estimates far away from 0 (with the direction depending on whether the effect is one of hypomethylation or hypermethylation, and the distance depending on the strength of the effect).

## RESULTS

All dimensions of data processing choices had some impact on downstream analysis results, in terms of the number (and identity) of differentially methylated probes, rankings of probes (by *p* value and logFC), estimates of logFC, and overall distribution of *p* values and logFC, corresponding to both real and spurious effects of interest. In some cases a consistent impact of a particular choice was observed, while in others there was no consistent pattern, or this pattern varied according to the other choices with which it was combined.

Tables 1–3 give information about the number of significantly differentially methylated probes in each of the models fitted for the combinations of filtering, normalisation, batch correction and cell adjustment approaches, for infant sex, maternal BMI (in the Standard Care group) and Tortoiseshell (in the Tabby group) respectively. Figures S1-S4 show the degree of overlap in the actual probes found to be significantly differently methylated between models for infant sex, intervention (at the mean BMI of the cohort), BMI (in the Standard Care group), and the effect of Short-Haired in the Tabby group. Figure 2 and Figs. 3–5 show the differences in ranking of probes (those which were in the top 10 in any model) by *p* value and log-Fold-Change for the same set of effects, and Tables S4–S6 gives Spearman Rank Correlation matrices for these rankings. The overall distribution of *p* values, and of log-Fold-Change estimates, for the same set of effects is shown in Figs. 6 and 7.

Below we discuss the effect of each dimension (probe filtering, normalisation, batch effects, cell type correction) on results.

### Effect of probe filtering pre-normalisation *vs* post-normalisation

Filtering probes prior to normalisation, compared to filtering after normalisation, led to modest differences in number of DMPs, rankings of probes by logFC and *p* value, and overall distributions of *p* values and logFC estimates. Filtering pre-normalisation produced different numbers of DMPs for infant sex, but the nature of the effect differed by normalisation method: in SWAN data there was a consistent pattern of fewer significant probes both negative and positive, while in BMIQ data there were fewer negative but more

**Table 1** Number of DMPs for infant sex (female), by probe filtering method, batch correction method, normalisation method and cell type method.

| Model | | | SQN | | BMIQ | | SWAN | |
|---|---|---|---|---|---|---|---|---|
| | | | Post Filtered | Pre Filtered | Post Filtered | Pre Filtered | Post Filtered | Pre Filtered |
| No ComBat | | | | | | | | |
| - Unadjusted | | Down | 15,088 | 14,878 | 7,935 | 7,554 | 7,581 | 6,890 |
| | | Up | 20,587 | 20,441 | 28,004 | 30,741 | 25,784 | 25,962 |
| - Adjusted for Batch | | Down | 13,132 | 13,215 | 6,602 | 6,112 | 7,100 | 6,140 |
| | | Up | 20,225 | 19,956 | 39,482 | 34,408 | 30,362 | 29,780 |
| - Adjusted for Batch + Cell | | Down | 28,406 | 28,633 | 15,855 | 14,900 | 14,239 | 10,408 |
| | | Up | 31,973 | 32,204 | 39,719 | 45,255 | 43,155 | 40,709 |
| Supervised ComBat | | | | | | | | |
| - Unadjusted[*] | | Down | 20,967 | 21,230 | 11,235 | 10,180 | 10,772 | 9,252 |
| | | Up | 25,690 | 25,320 | 41,518 | 48,191 | 45,193 | 43,237 |
| - Adjusted for Cell | | Down | 35,559 | 36,022 | 18,036 | 16,512 | 16,423 | 12,198 |
| | | Up | 37,068 | 36,972 | 56,521 | 64,851 | 68,769 | 65,109 |
| UnSupervised ComBat | | | | | | | | |
| - Unadjusted[*] | | Down | 14,603 | 14,763 | 7,634 | 6,836 | 7,344 | 6,377 |
| | | Up | 21,336 | 21,041 | 30,961 | 34,882 | 31,892 | 31,170 |
| - Adjusted for Cell | | Down | 28,012 | 28,286 | 14,060 | 12,916 | 13,030 | 9,560 |
| | | Up | 32,478 | 32,447 | 43,370 | 49,520 | 52,037 | 49,102 |

Notes.
 *No adjustment beyond the correction for batch as implemented in the ComBat algorithm.

**Table 2** DMPs for effect of maternal BMI in the standard care group.

| Model | SQN | | BMIQ | | SWAN | |
|---|---|---|---|---|---|---|
| | Post Filtered | Pre Filtered | Post Filtered | Pre Filtered | Post Filtered | Pre Filtered |
| No ComBat | | | | | | |
| - Unadjusted | 0 | 0 | 5 | 6 | 0 | 0 |
| - Adjusted for Batch | 0 | 0 | 6 | 0 | 0 | 0 |
| - Adjusted for Batch + Cell | 0 | 0 | 8 | 0 | 6 | 0 |
| Supervised ComBat | | | | | | |
| - Unadjusted | 0 | 0 | 0 | 10 | 0 | 0 |
| - Adjusted for Cell | 0 | 0 | 99 | 207 | 0 | 0 |
| UnSupervised ComBat | | | | | | |
| - Unadjusted | 0 | 0 | 0 | 0 | 0 | 0 |
| - Adjusted for Cell | 0 | 0 | 0 | 0 | 0 | 6 |

positive probes, and in SQN data there were more negative but fewer positive probes. In relation to effects of BMI, intervention, and fake groups, differences were harder to discern due to the lack of any DMPs for many models; however, when DMPs were present for an effect, there was a tendency for there to be a greater number of them in the pre-filtered data.

Probe rankings, by logFC and *p* value, tended to be relatively consistent between pre-filtered and post-filtered data, with some cases of larger discrepancies in rankings for individual probes. The discrepancies were more common, and larger, for fake group, BMI
**Table 3** DMPs for fake groups: 'short-haired' in 'Tabby'.

| Model | SQN | | BMIQ | | SWAN | |
|---|---|---|---|---|---|---|
| | Post Filtered | Pre Filtered | Post Filtered | Pre Filtered | Post Filtered | Pre Filtered |
| No ComBat | | | | | | |
| - Unadjusted | 0 | 0 | 0 | 0 | 0 | 0 |
| - Adjusted for Batch | 2,180 | 2,574 | 0 | 0 | 0 | 0 |
| - Adjusted for Batch + Cell | 0 | 0 | 0 | 0 | 0 | 0 |
| Supervised ComBat | | | | | | |
| - Unadjusted | 6,768 | 7,007 | 3 | 6 | 8 | 8 |
| - Adjusted for Cell | 123 | 133 | 1 | 1 | 0 | 0 |
| UnSupervised ComBat | | | | | | |
| - Unadjusted | 0 | 0 | 0 | 0 | 0 | 0 |
| - Adjusted for Cell | 0 | 0 | 0 | 0 | 0 | 0 |

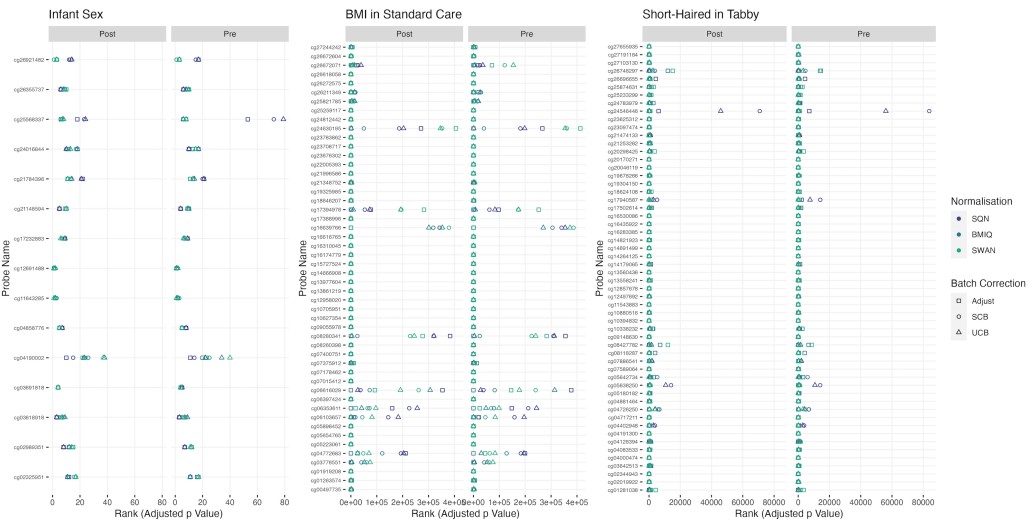

**Figure 2** **Probes ranked in top 10 by *p*-value in batch+cell adjusted model, for (A) infant sex, (B) BMI in standard care, (C) short-haired in Tabby.** For each probe the rank is given by pre- *vs* post-filtering, normalisation method, and batch-handling method. The model is one adjusting for batch (either explicitly in the model or *via* batch-correction algorithm) and cell type proportion. Adjust = adjusted for batch in the model; SCB = Supervised ComBat; UCB = Unsupervised ComBat.

and intervention effects than for infant sex. Similarly, there were no dramatic differences in distributions of *p* values or logFC estimates for infant sex; there were differences in distribution between pre- and post-filtered data for fake group, intervention and BMI effects, but there was no consistent pattern to these differences.

The question of whether probe filtering should be carried out before or after normalisation is one which has received surprisingly little attention in the literature, but our results suggest that it can make a difference to findings in some contexts. In particular there may be a higher risk of spurious findings in pre-filtered data, but there may also be a risk of failing to detect true differences –either any differences, or specifically

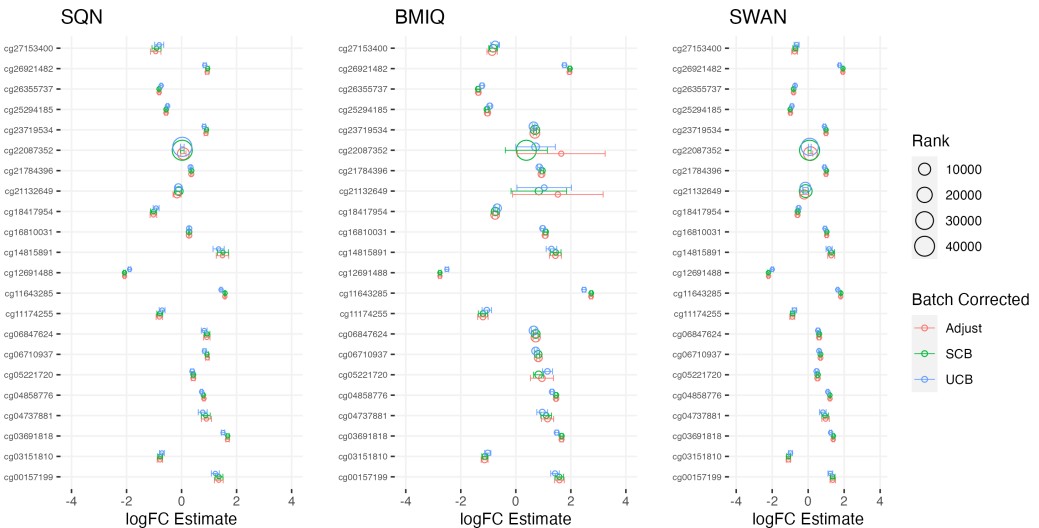

**Figure 3 Top 10 probes by LogFC: infant sex.** Largest LogFC for Infant Sex (female), by normalisation and batch correction method.

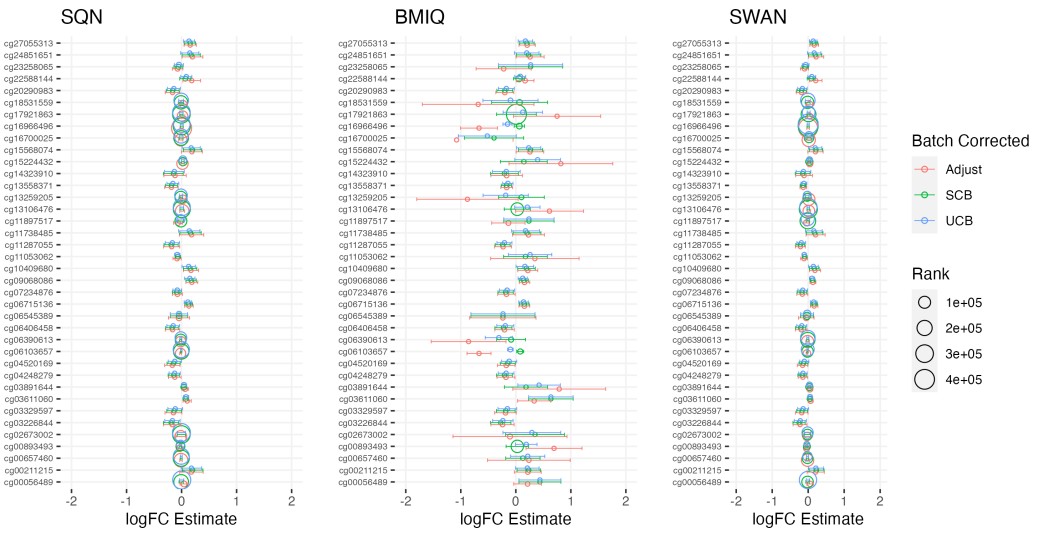

**Figure 4 Top 10 probes by LogFC: BMI in standard care.** Largest LogFC for effect of BMI in standard care group, by normalisation and batch correction method.

hypomethylated or hypermethylated loci, depending upon the normalisation method employed.

## Effect of normalisation method

Normalisation method had a substantial influence on number and identity of DMPs, rankings of probes and *p* values, and distributions of *p* values and logFC estimates. For infant sex, SQN data consistently had the highest number of significant negative probes and the lowest number of significant positive probes, while SWAN data always had the lowest

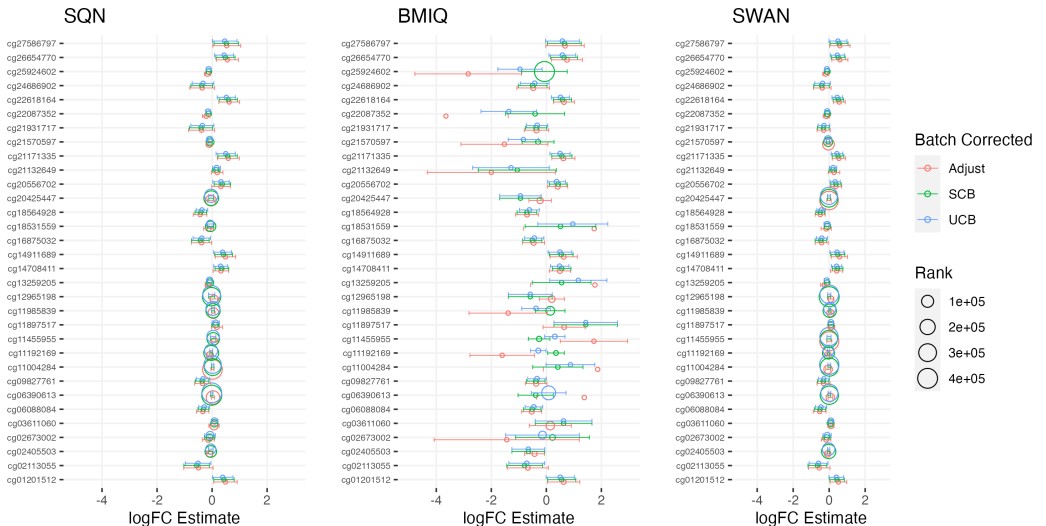

**Figure 5** **Top 10 probes by LogFC: 'short-haired' in 'Tabby'.** Largest LogFC for effect of 'Short-Haired' in 'Tabby' group, by normalisation and batch correction method.

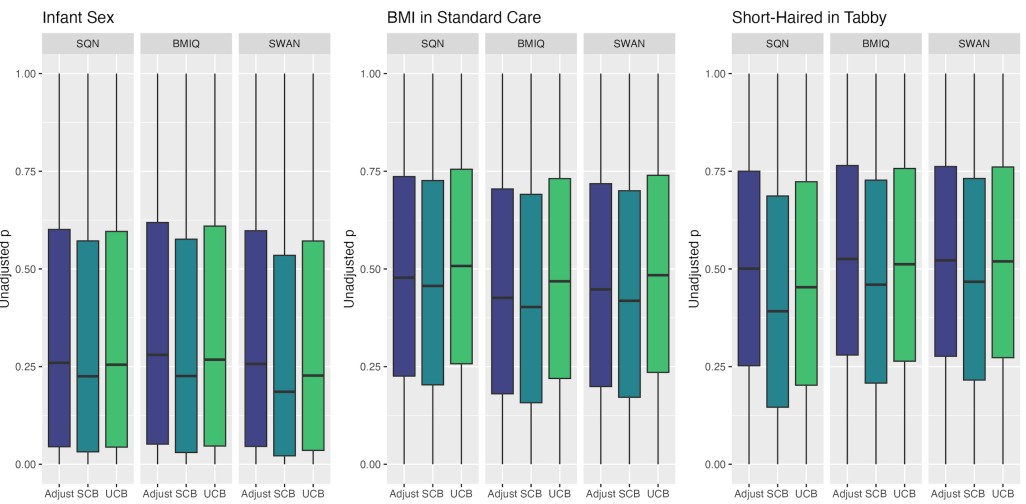

**Figure 6** **Distribution of unadjusted P values by normalisation and batch correction method, for batch and cell adjusted models.** Only models from data where probe filtering was performed post-normalisation are included. The model is one adjusting for batch (either explicitly in the model or *via* batch-correction algorithm) and cell type proportion.

number of significant positive probes. For BMI and intervention effects, only BMIQ data produced DMPs where no ComBat processing was used; in data processed using supervised ComBat, all three normalisation methods resulted in some DMPs, but the number and identity of these probes differed. In fake group data, SQN data produced a large number of significant probes in non-ComBat-processed and supervised-ComBat data, while BMIQ and SWAN data produced a small number of probes in supervised-ComBat data only;

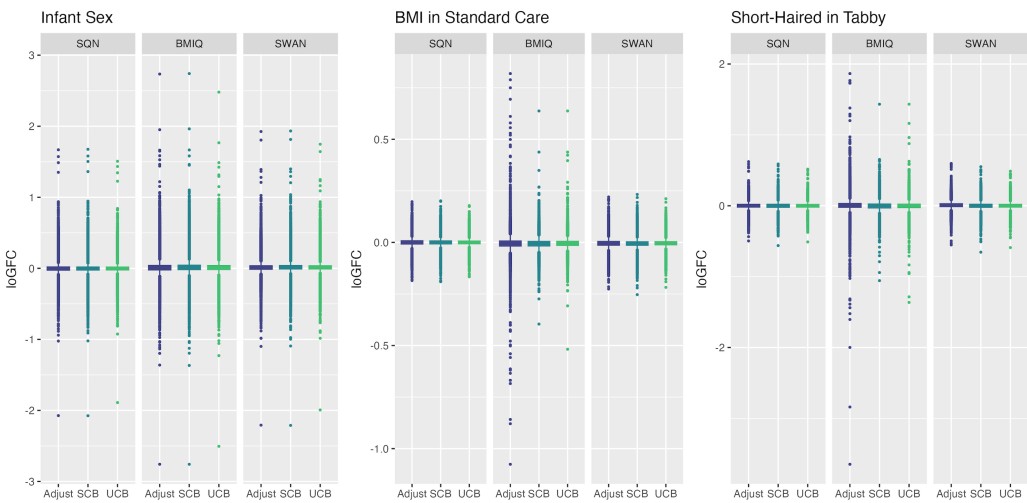

**Figure 7  Distribution of log-fold-change estimates by normalisation and batch correction method, for batch and cell adjusted models.** Only models from data where probe filtering was performed post-normalisation are included. The model is one adjusting for batch (either explicitly in the model or *via* batch-correction algorithm) and cell type proportion.

again, the number of significant probes differed between the normalisation methods (see Tables 1–3 and Figs. S1–S4).

There was a fair degree of consistency in rankings of probes by *p* value for infant sex, but some large discrepancies in rankings for BMI, intervention, and fake group effects. The rankings were less consistent for highest-ranked probes by logFC, with some quite large differences in both rankings and effect estimates (including different directions of effect) for infant sex, BMI, intervention and fake groups. BMIQ estimates tended to be more extreme (further from 0) than the other two methods.

Distributions of *p* values and logFC estimates also differed between normalisation methods. For *p* values the differences were not consistent across models and effects, but for logFC there was a clear difference between BMIQ and the other two methods, with the range of estimates in BMIQ data being much more widely dispersed; SQN and SWAN data had more similar distributions, but SQN was moderately narrower than SWAN across all effects and models.

Overall, there was little difference between SQN and SWAN methods when adjusting for batch in the model. There is some evidence that SQN would result in fewer significant DMPs than SWAN for known effects (particularly when using supervised ComBat), but (many) more spuriously significant DMPs than either SWAN or BMIQ where effects are absent. The behaviour of BMIQ was more variable depending on other dimensions of the pipeline, but had a wider dispersion of logFC estimates than the other methods, particularly when adjusting for batch in the model. This tended to result in more DMPs in some scenarios, but in general will lead effect estimates derived from BMIQ data to be more extreme (and probably overestimates of the true effect).

## Effect of batch correction method

There were clear differences in all dimensions between batch correction methods. For all effects (infant sex, BMI, intervention and fake groups), supervised ComBat processing produced a larger number of DMPs compared to either unsupervised ComBat processing or adjustment for batch in the analysis model. The difference between unsupervised ComBat and batch-adjustment was less consistent for infant sex effects, but for BMI, intervention and fake group effects, there were no DMPs in unsupervised ComBat models, whereas there were a few for batch-adjusted models.

Rankings of top probes by $p$ value were relatively consistent between batch-adjustment methods for infant sex, but there were some large discrepancies particularly for BMI and intervention effects, and especially in BMIQ data. The same phenomenon was observed for logFC rankings, which also showed a tendency for logFC estimates in unsupervised-ComBat data to be smaller in absolute magnitude (closer to 0).

The distribution of $p$ values showed clear and consistent differences between batch-correction methods, with the distribution in supervised ComBat data shifted substantially towards 0 relative to both unsupervised ComBat and batch-adjusted models, for all effects. For logFC estimates, supervised ComBat and batch-adjusted data were generally fairly similar, but unsupervised ComBat data generally resulted in a narrower range. This means that supervised ComBat will tend to produce more statistically significant probes, regardless of the presence or absence of an effect. Conversely, effect estimates from unsupervised ComBat may be underestimated; at least, they will tend to be smaller in magnitude than those derived from data where batch is handled differently.

Of particular note is the combination of SQN normalisation and either adjustment for batch in the model, or use of supervised ComBat. These combinations produced an extremely large number of significant DMPs for fake group effects; this was more extreme in the case of supervised ComBat (producing over 6,000 DMPs) than when adjusting for batch (somewhat over 2,000 DMPs). Additional adjustment for cell type proportion ameliorated this effect, as discussed below, but in the case of supervised ComBat data, did not eliminate spurious findings. This suggests that batch adjustment may be particularly ill-advised in the context of SQN normalisation; since SQN involves between-array as well as within-array normalisation, additional adjustment for batch may be over-correcting.

## Effect of adjustment for estimated cell type proportion

Adjustment for cell type proportion affected results, but the impact was not consistent across the different types of effects studied. Adjustment for batch resulted in a substantially larger number of DMPs (both negative and positive) for infant sex, but reduced the number of DMPs for fake groups (for models where there were DMPs for fake groups effects). For BMI and intervention, the effect of cell type adjustment was mostly but not entirely to produce more DMPs.

The effect of cell type adjustment on top probe rankings was fairly modest, although some quite large discrepancies were observed for $p$ value rankings, logFC rankings, and logFC estimates. The effect on distribution of $p$ values depended on the effect: for infant sex, adjustment for cell type proportion consistently (for all normalisation and batch-correction
methods) shifted the distribution downwards towards 0 (*i.e.,* more statistically significant probes), whereas the differences were less consistent and smaller in BMI, intervention and fake group effects. There were no large or consistent differences in distribution of logFC estimates between cell-type-adjusted and non-adjusted models.

Overall, adjustment for cell type proportion tended to improve model behaviour regarding spurious results: the number of significantly differentially methylated probes decreased with adjustment for cell type proportion, though they were not always eliminated. The number of differentially methylated probes for infant sex was increased, which may reflect either improvement (greater ability to detect true effects due to removal of noise due to cell type differences) or harm (greater number of spurious effects) depending on whether the extra probes are in fact differentially methylated between males and females; without knowing the true number and identity of DMPs, we cannot be certain. Similarly, adjustment for cell type proportion increased the number of DMPs for BMI effects in BMIQ and SWAN data, in one scenario (BMIQ with Supervised ComBat) by a substantial amount (from 99 to 2017 DMPs) and these are most likely to be false positives.

## DISCUSSION

Different choices in probe filtering, normalisation, batch handling, and adjustment for cell types resulted in different findings regarding the presence and identity of differentially methylated probes, rankings of probes by *p* value and log-fold-change, and different overall distributions of *p* values and log-fold-change estimates. While some of these differences were relatively modest, our results nevertheless show that particular combinations of data processing and analysis choices may result in spurious false positive findings, and/or potentially the failure to detect true effects. Additionally, while the magnitude of effect estimates is often not considered in differential methylation studies, some pipelines may result in an overestimate or underestimate of the true effect. Importantly, the results tended to depend on *combinations* of choices rather than individual elements of the analysis pipeline.

The results of our analyses are consistent with other investigations which have been undertaken into different data-processing and analysis choices. As noted above, the potential for 'false positives' to result from supervised batch-correction methods specifying effects of interest has been previously identified by a number of authors (*Nygaard, Rødland & Hovig, 2016a*; *Price & Robinson, 2018*; *Zindler et al., 2020*). Our finding that the distribution of *p* values in the supervised ComBat algorithm tends to shift the *p* value distribution downward is consistent with the finding of *Nygaard, Rødland & Hovig (2016a)* that, in contexts where the effects of interest are not evenly spread between batches, the distribution of F-statistics will be biased upwards. While implementation as an unsupervised method may be preferable, our findings suggest that this may create a different problem, with the estimates of log-fold-change corresponding to effects of interest biased towards zero.

*Wu et al. (2014)* compared a variety of normalisation approaches noted a tendency for more statistically significant differences to arise in SQN data, which they hypothesise
may be due to reduced overall variance. In our investigation, the main context in which SQN data produced a large number of spurious differentially methylated probes was when supervised ComBat, or adjustment for batch in the model, was used; we additionally found that adjustment for cell type proportion reduced the number of spurious findings (while not necessarily eliminating them). Thus, SQN is not universally more prone to producing spurious findings than other normalisation methods.

Our findings do not suggest that there is one particular combination of methods which can be guaranteed to work in all contexts, and some of the recommendations which have been made by others may need to be modified somewhat. For example *Nygaard, Rødland & Hovig (2016b)* conclude that adjustment for batch in the model is preferable to the use of batch-correction algorithms, but our results suggest that this is inadvisable for data that have been normalised using SQN; in our data, this combination resulted in a large number of spurious findings. In general, while our results support others' findings that supervised bach-correction algorithms should not be used, there does not appear to be much difference between unsupervised batch-correction and adjustment for batch in the model. The only caveat here is that some of our results (particularly regarding effects of infant sex) suggest that unsupervised ComBat may underestimate the magnitude of effects, as the distribution of logFC estimates was substantially narrower than other methods. The use of a more stringent method of Type I error control may also help to reduce the number of spurious findings: the use of FDR correction methods such as Benjamini–Hochberg, while very common (*Maksimovic, Phipson & Oshlack, 2017*), may not be sufficient to deal with higher rates of spurious results (*Nygaard, Rødland & Hovig, 2016a*). In our data, the use of the Holm method (which controls the Family-Wise Error Rate) reduced, but did not eliminate, spurious findings associated with fake group effects. Investigation of DNA regions, rather than probe-wise analysis, may also help to differentiate true methylation differences from spurious ones (*Wang et al., 2015*): the statistically significant DMPs for fake group effects (as well as for BMI and intervention effects) tended to be isolated rather than being grouped in the same region, and in our companion paper, we found no significant differences in methylation for groups of probes on candidate genes (*Louise et al., 2022*).

One limitation of our study is our inability to compare model behaviour in relation to known effects. It was relatively simple to create fake groups to study behaviour of models for effects which were known *not* to exist, but as we do not know the truth about which effects actually exist in our data, we could not compare behaviour of models in their ability to detect these known effects. Simulated data could potentially be used for this purpose; however, the effects in the simulated data would have to be biologically plausible. This was beyond the scope of our study; however, it is a good subject for future research. We used infant sex as the nearest proxy to a known effect, as we knew at least that some effects existed. However, we cannot say whether, and to what extent, the differences observed in relation to infant sex reflect spurious findings *versus* the failure to detect true effects.

Overall, as many other authors have noted, researchers working with DNAm data should better understand the methods built into standard pipelines (*Price & Robinson, 2018*; *Zindler et al., 2020*), and should better document the specific data-processing methods

used (*Nygaard, Rødland & Hovig, 2016a*; *Zindler et al., 2020*). It is also important, in our view, to pay more attention to the context in which a particular epigenome-wide analysis is performed. For example, a less stringent method of Type I error control may often be chosen because the study is exploratory (hypothesis-generating) rather than confirmatory, and it is considered more important not to miss potential findings than to rule out spurious ones. In this case, the results from such studies should be interpreted accordingly: as suggestive findings which cannot be confidently accepted until they are validated in new data. The validation of existing findings should be treated as a high priority in epigenetics research (*Price & Robinson, 2018*).

Additionally, the degree of confidence that can be placed in any new discoveries could be enhanced by performing sensitivity analyses—re-performing analyses using different normalisation methods, batch correction methods, or models—which we believe should become standard in this area.

## ACKNOWLEDGEMENTS

We would like to acknowledge the participants in the LIMIT Randomised Controlled Trial, whose samples and data are used in this study, as well as the LIMIT research team who collected data and samples for the study. We would also like to acknowledge Shobana Navaneethabalakrishnan, the Academic Editor at PeerJ and two anonymous reviewers, whose helpful suggestions have substantially improved the manuscript.

### Funding

The LIMIT Randomised Trial was funded by an NHMRC grant (ID519240), awarded to Jodie M. Dodd. Funding for the DNA methylation analysis was from the Commission of the European Communities, the 7th Framework Programme, contract FP7-289346-EARLY NUTRITION. Jodie M. Dodd was also supported by NHMRC Practitioner Fellowships (ID627005 and ID1078980) and Investigator Grant (ID1196133). The funders had no role in study design, data collection and analysis, decision to publish, or preparation of the manuscript.

### Grant Disclosures

The following grant information was disclosed by the authors:
NHMRC: ID519240.
Commission of the European Communities.
7th Framework Programme: FP7-289346-EARLY NUTRITION.
NHMRC Practitioner Fellowships: ID627005, ID1078980.
Investigator Grant: ID1196133.

### Competing Interests

The authors declare there are no competing interests.

## Author Contributions

- Jennie Louise conceived and designed the experiments, performed the experiments, analyzed the data, prepared figures and/or tables, authored or reviewed drafts of the article, and approved the final draft.
- Andrea R. Deussen conceived and designed the experiments, performed the experiments, prepared figures and/or tables, authored or reviewed drafts of the article, and approved the final draft.
- Jodie M. Dodd conceived and designed the experiments, performed the experiments, authored or reviewed drafts of the article, and approved the final draft.

## Human Ethics

The following information was supplied relating to ethical approvals (i.e., approving body and any reference numbers):

The study was reviewed by the ethics committee of each participating institution: the Women's and Children's Health Network Human Research Ethics Committee (1839 & 2051), the Central and Northern Adelaide Health Network Human Research Ethics Committee (2008033) and the Southern Adelaide Local Health Network Human Research Ethics Committee (128/08).

## Data Availability

The R scripts contain all code for data processing and analysis are available in the Supplementary File.

## Supplemental Information

Supplemental information for this article can be found online at http://dx.doi.org/10.7717/peerj.14786#supplemental-information.

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
