# Peer review of "Data processing choices can affect findings in differential methylation analyses: an investigation using data from the LIMIT RCT"

_PeerJ, doi:10.7717/peerj.14786_

## Round 0.1 · original submission · Minor Revisions

The authors are asked to provide images with high resolutions. The introduction needs elaboration of previous studies with references and how those studies helped to find the lacunae and how the current study addressed them.

Reviewer 1 ·

Basic reporting

This paper is well and clearly written. The background and introduction are sufficiently provided. And the paper is well-structured and delivered.

Experimental design

This paper sought to perform a systematic evaluation of data processing through differential methylation analyses of DNA methylation array data. I think this is a timely and important question for the field and certainly will fit within the scope of PeerJ. The methods were clearly described and the analyses were well performed. I only have one minor suggestion here: The authors might want to consider providing the source scripts/codes of their analyses pipeline.

Validity of the findings

Conclusions are well articulated and supported by the analyses. But I do suggest the authors could improve their image quality of figures. Right now it is hard to see the labels clearly.

Reviewer 2 ·

Basic reporting

The paper addresses how data processing choices can impact the findings of any differential methylation analyses. In an area where a wide range of methods and tools to be used for different data processing steps, dealing with these set of questions is inevitable, hence the paper succeeds in tackling a very pressing and relevant challenge that faces anyone studying DNA methylation. The authors used a clear and professional English throughout the manuscript (However, I noticed an excessive use of the "," in a way that sometimes come as redundant and confusing).
The manuscripts offered a range of literature references relevant to the questions subject to study. Even though the manuscript referenced a list of existing methods and some existing evaluation papers, the literature review section lacked a detailed discussion of what is already established from previous work and how this new manuscript build on it. The authors targeted that in the discussion section, but I would recommend moving part of that discussion to the literature section. That would strengthen the motivation for their study and answers why the questions they are trying to investigate haven't been answered already by the referenced papers.
The structure and story of the manuscript was clear and well organized. However I found most figures hard to read because of resolution or scaling issues. I left suggestions in the annotated pdf of how that can be improved.

Experimental design

I found the experimental design clear in most of the manuscript (except a few places where I left comments in the pdf). The primary research question was well defined and very relevant to the scope of the journal. The authors investigate that question in the scope of a cord blood methylation dataset from their previous work. Dataset appeared to be well designed and suited to answer their questions. My biggest concern was regarding their evaluation approach. The authors chose five dimensions on which they built their evaluation of the different data processing choices. However there were no validation step nor any baseline that can form as a ground truth that can be used as a reference to be able to derive strong conclusions. Most conclusions, in my opinion, appeared to be heavily relying on the number of identified DMPs among the five criteria. Whereas looking at number with no baseline to distinguish between "real" and "spurious" findings could be misleading. Other than my annotated suggestions throughout the pdf, I would highly encourage the authors to include some sort of validation step to add solid foundation to their conclusions.

Validity of the findings

In the discussion section, the authors added a clear contrast between their findings and the findings from previous work, which was very important to the scope of the manuscript and linked their investigation to the literature. Supplementary tables showed some detailed data of their findings but I didn't see access to the raw data itself. That is important to be provided for reproducibility. If it is provided in the accompanying paper, I would still recommend adding a link to how to access the raw data itself to this manuscript.
Most of the key findings came in the discussion section rather than their corresponding subsections under results. That caused, in some cases, these conclusions vague and weakly connected to the study and how they were derived. I would suggest moving stating each of these conclusions into their corresponding subsections with the details of how each is derived from the observed results, while keeping their mention in the discussion section brief as takeaway messages.

Additional comments

The manuscript started strong with an exciting questions, clearly communicated message and well-written text. But I found the results section fairly underwhelming considering the importance and difficulty of the research questions.
Even though I recommend the article to be accepted with revisions, I strongly encourage the authors to consider my suggestions especially in the results section. It will boost the article strength and value in the literature.

Annotated reviews are not available for download in order to protect the identity of reviewers who chose to remain anonymous.

---

## Round 0.2 · accepted · Accept

The authors have addressed all the comments and the manuscript is ready for publication.